# Winner-Take-All Autoencoders

**Alireza Makhzani, Brendan Frey**
University of Toronto
`makhzani, frey@psi.toronto.edu`

## Abstract

In this paper, we propose a winner-take-all method for learning hierarchical sparse representations in an unsupervised fashion. We first introduce fully-connected winner-take-all autoencoders which use mini-batch statistics to directly enforce a lifetime sparsity in the activations of the hidden units. We then propose the convolutional winner-take-all autoencoder which combines the benefits of convolutional architectures and autoencoders for learning shift-invariant sparse representations. We describe a way to train convolutional autoencoders layer by layer, where in addition to lifetime sparsity, a spatial sparsity within each feature map is achieved using winner-take-all activation functions. We will show that winner-take-all autoencoders can be used to to learn deep sparse representations from the MNIST, CIFAR-10, ImageNet, Street View House Numbers and Toronto Face datasets, and achieve competitive classification performance.

## 1 Introduction

Recently, supervised learning has been developed and used successfully to produce representations that have enabled leaps forward in classification accuracy for several tasks [1]. However, the question that has remained unanswered is whether it is possible to learn as "powerful" representations from unlabeled data without any supervision. It is still widely recognized that unsupervised learning algorithms that can extract useful features are needed for solving problems with limited label information. In this work, we exploit sparsity as a generic prior on the representations for unsupervised feature learning. We first introduce the fully-connected winner-take-all autoencoders that learn to do sparse coding by directly enforcing a winner-take-all *lifetime* sparsity constraint. We then introduce convolutional winner-take-all autoencoders that learn to do shift-invariant/convolutional sparse coding by directly enforcing winner-take-all *spatial* and *lifetime* sparsity constraints.

## 2 Fully-Connected Winner-Take-All Autoencoders

Training sparse autoencoders has been well studied in the literature. For example, in [2], a "lifetime sparsity" penalty function proportional to the KL divergence between the hidden unit marginals ($\hat{\rho}$) and the target sparsity probability ($\rho$) is added to the cost function: $\lambda \mathrm{KL}(\rho \| \hat{\rho})$. A major drawback of this approach is that it only works for certain target sparsities and is often very difficult to find the right $\lambda$ parameter that results in a properly trained sparse autoencoder. Also KL divergence was originally proposed for sigmoidal autoencoders, and it is not clear how it can be applied to ReLU autoencoders where $\hat{\rho}$ could be larger than one (in which case the KL divergence can not be evaluated). In this paper, we propose Fully-Connected Winner-Take-All (FC-WTA) autoencoders to address these concerns. FC-WTA autoencoders can aim for any target sparsity rate, train very fast (marginally slower than a standard autoencoder), have no hyper-parameter to be tuned (except the target sparsity rate) and efficiently train all the dictionary atoms even when very aggressive sparsity rates (*e.g.*, 1%) are enforced.

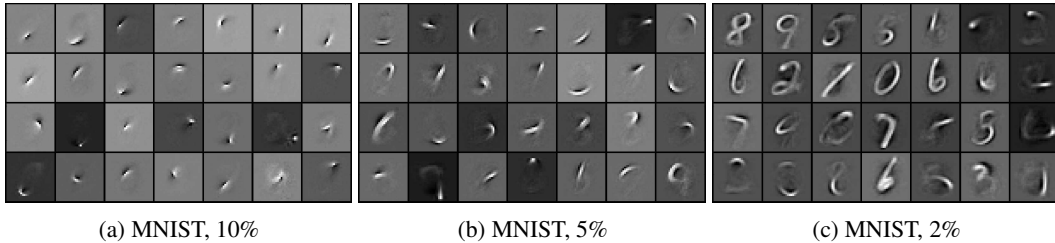

|(a) MNIST, 10%|(b) MNIST, 5%|(c) MNIST, 2%|

Figure 1: Learnt dictionary (decoder) of FC-WTA with 1000 hidden units trained on MNIST

Sparse coding algorithms typically comprise two steps: a highly non-linear sparse encoding operation that finds the "right" atoms in the dictionary, and a linear decoding stage that reconstructs the input with the selected atoms and update the dictionary. The FC-WTA autoencoder is a non-symmetric autoencoder where the encoding stage is typically a stack of several ReLU layers and the decoder is just a linear layer. In the feedforward phase, after computing the hidden codes of the last layer of the encoder, rather than reconstructing the input from all of the hidden units, for each hidden unit, we impose a lifetime sparsity by keeping the $k$ percent largest activation of that hidden unit across the mini-batch samples and setting the rest of activations of that hidden unit to zero. In the backpropagation phase, we only backpropagate the error through the $k$ percent non-zero activations. In other words, we are using the min-batch statistics to approximate the statistics of the activation of a particular hidden unit across all the samples, and finding a hard threshold value for which we can achieve $k\%$ lifetime sparsity rate. In this setting, the highly nonlinear encoder of the network (ReLUs followed by top-k sparsity) learns to do sparse encoding, and the decoder of the network reconstructs the input linearly. At test time, we turn off the sparsity constraint and the output of the deep ReLU network will be the final representation of the input. In order to train a stacked FC-WTA autoencoder, we fix the weights and train another FC-WTA autoencoder on top of the fixed representation of the previous network.

The learnt dictionary of a FC-WTA autoencoder trained on MNIST, CIFAR-10 and Toronto Face datasets are visualized in Fig. 1 and Fig 2. For large sparsity levels, the algorithm tends to learn very local features that are too primitive to be used for classification (Fig. 1a). As we decrease the sparsity level, the network learns more useful features (longer digit strokes) and achieves better classification (Fig. 1b). Nevertheless, forcing too much sparsity results in features that are too global and do not factor the input into parts (Fig. 1c). Section 4.1 reports the classification results.

**Winner-Take-All RBMs.** Besides autoencoders, WTA activations can also be used in Restricted Boltzmann Machines (RBM) to learn sparse representations. Suppose $\mathbf{h}$ and $\mathbf{v}$ denote the hidden and visible units of RBMs. For training WTA-RBMs, in the positive phase of the contrastive divergence, instead of sampling from $P(h_i|\mathbf{v})$, we first keep the $k\%$ largest $P(h_i|\mathbf{v})$ for each $h_i$ across the mini-batch dimension and set the rest of $P(h_i|\mathbf{v})$ values to zero, and then sample $h_i$ according to the sparsified $P(h_i|\mathbf{v})$. Filters of a WTA-RBM trained on MNIST are visualized in Fig. 3. We can see WTA-RBMs learn longer digit strokes on MNIST, which as will be shown in Section 4.1, improves the classification rate. Note that the sparsity rate of WTA-RBMs (*e.g.*, 30%) should not be as aggressive as WTA autoencoders (*e.g.*, 5%), since RBMs are already being regularized by having binary hidden states.

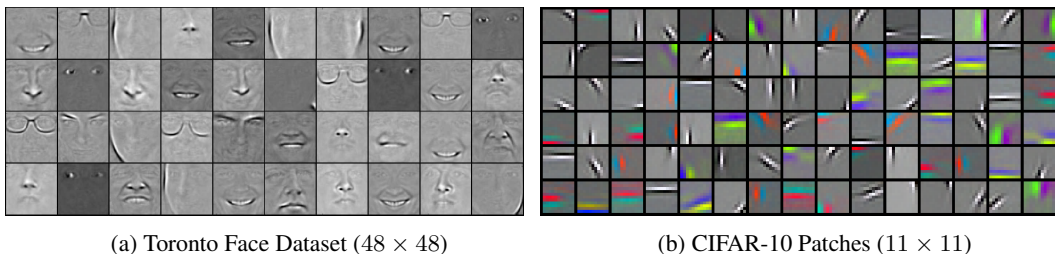

|(a) Toronto Face Dataset ($48 \times 48$)|(b) CIFAR-10 Patches ($11 \times 11$)|

Figure 2: Dictionaries (decoder) of FC-WTA autoencoder with 256 hidden units and sparsity of 5%

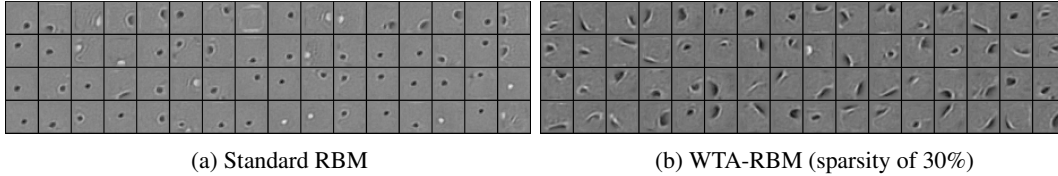

| (a) Standard RBM | (b) WTA-RBM (sparsity of 30%) |

Figure 3: Features learned on MNIST by 256 hidden unit RBMs.

## 3 Convolutional Winner-Take-All Autoencoders

There are several problems with applying conventional sparse coding methods on large images. First, it is not practical to directly apply a fully-connected sparse coding algorithm on high-resolution (*e.g.*, $256 \times 256$) images. Second, even if we could do that, we would learn a very redundant dictionary whose atoms are just shifted copies of each other. For example, in Fig. 2a, the FC-WTA autoencoder has allocated different filters for the same patterns (*i.e.*, mouths/noses/glasses/face borders) occurring at different locations. One way to address this problem is to extract random image patches from input images and then train an unsupervised learning algorithm on these patches in isolation [3]. Once training is complete, the filters can be used in a convolutional fashion to obtain representations of images. As discussed in [3, 4], the main problem with this approach is that if the receptive field is small, this method will not capture relevant features (imagine the extreme of $1 \times 1$ patches). Increasing the receptive field size is problematic, because then a very large number of features are needed to account for all the position-specific variations within the receptive field. For example, we see that in Fig. 2b, the FC-WTA autoencoder allocates different filters to represent the same horizontal edge appearing at different locations within the receptive field. As a result, the learnt features are essentially shifted versions of each other, which results in redundancy between filters. Unsupervised methods that make use of convolutional architectures can be used to address this problem, including convolutional RBMs [5], convolutional DBNs [6, 5], deconvolutional networks [7] and convolutional predictive sparse decomposition (PSD) [4, 8]. These methods learn features from the entire image in a convolutional fashion. In this setting, the filters can focus on learning the shapes (*i.e.*, "what"), because the location information (*i.e.*, "where") is encoded into feature maps and thus the redundancy among the filters is reduced.

In this section, we propose Convolutional Winner-Take-All (CONV-WTA) autoencoders that learn to do shift-invariant/convolutional sparse coding by directly enforcing winner-take-all *spatial* and *lifetime* sparsity constraints. Our work is similar in spirit to deconvolutional networks [7] and convolutional PSD [4, 8], but whereas the approach in that work is to break apart the recognition pathway and data generation pathway, but learn them so that they are consistent, we describe a technique for directly learning a sparse convolutional autoencoder.

A shallow convolutional autoencoder maps an input vector to a set of feature maps in a convolutional fashion. We assume that the boundaries of the input image are zero-padded, so that each feature map has the same size as the input. The hidden representation is then mapped linearly to the output using a deconvolution operation (Appendix A.1). The parameters are optimized to minimize the mean square error. A non-regularized convolutional autoencoder learns useless delta function filters that copy the input image to the feature maps and copy back the feature maps to the output. Interestingly, we have observed that even in the presence of denoising[9]/dropout[10] regularizations, convolutional autoencoders still learn useless delta functions. Fig. 4a depicts the filters of a convolutional autoencoder with 16 maps, 20% input and 50% hidden unit dropout trained on Street View House Numbers dataset [11]. We see that the 16 learnt delta functions make 16 copies of the input pixels, so even if half of the hidden units get dropped during training, the network can still rely on the non-dropped copies to reconstruct the input. This highlights the need for new and more aggressive regularization techniques for convolutional autoencoders.

The proposed architecture for CONV-WTA autoencoder is depicted in Fig. 4b. The CONV-WTA autoencoder is a non-symmetric autoencoder where the encoder typically consists of a stack of several ReLU convolutional layers (*e.g.*, $5 \times 5$ filters) and the decoder is a linear deconvolutional layer of larger size (*e.g.*, $11 \times 11$ filters). We chose to use a deep encoder with smaller filters (*e.g.*, $5 \times 5$) instead of a shallow one with larger filters (*e.g.*, $11 \times 11$), because the former introduces more

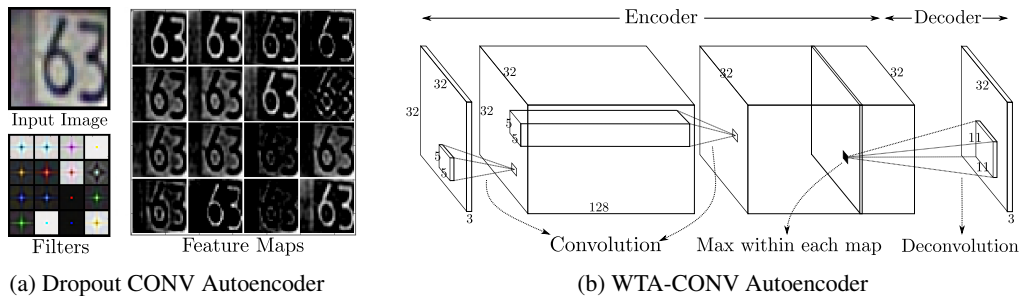

(a) Dropout CONV Autoencoder          (b) WTA-CONV Autoencoder

Figure 4: (a) Filters and feature maps of a denoising/dropout convolutional autoencoder, which learns useless delta functions. (b) Proposed architecture for CONV-WTA autoencoder with spatial sparsity (128conv5-128conv5-128deconv11).

non-linearity and regularizes the network by forcing it to have a decomposition over large receptive fields through smaller filters. The CONV-WTA autoencoder is trained under two winner-take-all sparsity constraints: *spatial sparsity* and *lifetime sparsity*.

## 3.1 Spatial Sparsity

In the feedforward phase, after computing the last feature maps of the encoder, rather than reconstructing the input from all of the hidden units of the feature maps, we identify the single largest hidden activity within each feature map, and set the rest of the activities as well as their derivatives to zero. This results in a sparse representation whose sparsity level is the number of feature maps. The decoder then reconstructs the output using only the active hidden units in the feature maps and the reconstruction error is only backpropagated through these hidden units as well.

Consistent with other representation learning approaches such as triangle $k$-means [3] and deconvolutional networks [7, 12], we observed that using a softer sparsity constraint at test time results in a better classification performance. So, in the CONV-WTA autoencoder, in order to find the final representation of the input image, we simply turn off the sparsity regularizer and use ReLU convolutions to compute the last layer feature maps of the encoder. After that, we apply max-pooling (*e.g.*, over $4 \times 4$ regions) on these feature maps and use this representation for classification tasks or in training stacked CONV-WTA as will be discussed in Section 3.3. Fig. 5 shows a CONV-WTA autoencoder that was trained on MNIST.

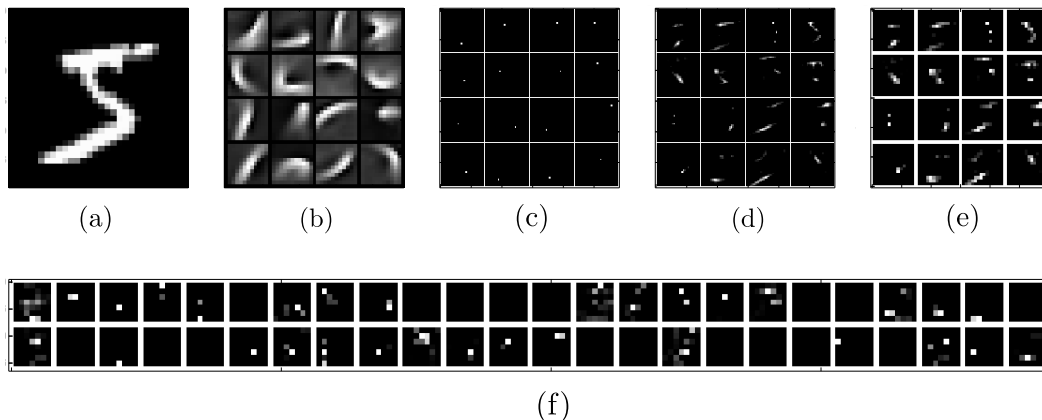

(a)          (b)          (c)          (d)          (e)

(f)

Figure 5: The CONV-WTA autoencoder with 16 first layer filters and 128 second layer filters trained on MNIST: (a) Input image. (b) Learnt dictionary (deconvolution filters). (c) 16 feature maps while training (spatial sparsity applied). (d) 16 feature maps after training (spatial sparsity turned off). (e) 16 feature maps of the first layer after applying local max-pooling. (f) 48 out of 128 feature maps of the second layer after turning off the sparsity and applying local max-pooling (final representation).

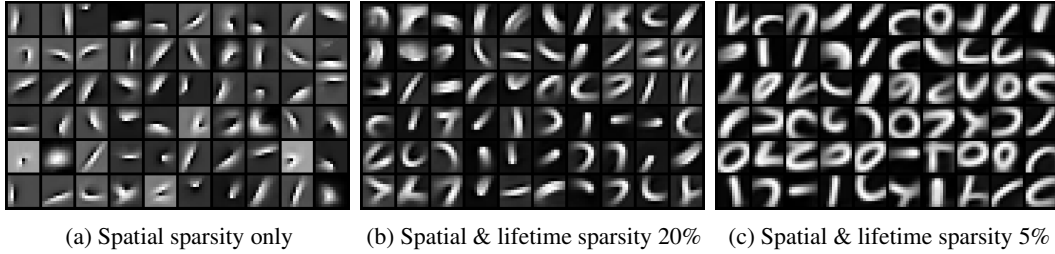

| (a) Spatial sparsity only | (b) Spatial & lifetime sparsity 20% | (c) Spatial & lifetime sparsity 5% |

Figure 6: Learnt dictionary (deconvolution filters) of CONV-WTA autoencoder trained on MNIST (64conv5-64conv5-64conv5-64deconv11).

### 3.2 Lifetime Sparsity

Although spatial sparsity is very effective in regularizing the autoencoder, it requires all the dictionary atoms to contribute in the reconstruction of every image. We can further increase the sparsity by exploiting the winner-take-all *lifetime* sparsity as follows. Suppose we have 128 feature maps and the mini-batch size is 100. After applying spatial sparsity, for each filter we will have 100 "winner" hidden units corresponding to the 100 mini-batch images. During feedforward phase, for each filter, we only keep the $k\%$ largest of these 100 values and set the rest of activations to zero. Note that despite this aggressive sparsity, every filter is forced to get updated upon visiting every mini-batch, which is crucial for avoiding the dead filter problem that often occurs in sparse coding.

Fig. 6 and Fig. 7 show the effect of the lifetime sparsity on the dictionaries trained on MNIST and Toronto Face dataset. We see that similar to the FC-WTA autoencoders, by tuning the lifetime sparsity of CONV-WTA autoencoders, we can aim for different sparsity rates. If no lifetime sparsity is enforced, we learn local filters that contribute to every training point (Fig. 6a and 7a). As we increase the lifetime sparsity, we can learn rare but useful features that result in better classification (Fig. 6b). Nevertheless, forcing too much lifetime sparsity will result in features that are too diverse and rare and do not properly factor the input into parts (Fig. 6c and 7b).

### 3.3 Stacked CONV-WTA Autoencoders

The CONV-WTA autoencoder can be used as a building block to form a hierarchy. In order to train the hierarchical model, we first train a CONV-WTA autoencoder on the input images. Then we pass all the training examples through the network and obtain their representations (last layer of the encoder after turning off sparsity and applying local max-pooling). Now we treat these representations as a new dataset and train another CONV-WTA autoencoder to obtain the stacked representations. Fig. 5(f) shows the deep feature maps of a stacked CONV-WTA that was trained on MNIST.

### 3.4 Scaling CONV-WTA Autoencoders to Large Images

The goal of convolutional sparse coding is to learn *shift-invariant* dictionary atoms and encoding filters. Once the filters are learnt, they can be applied convolutionally to any image of any size, and produce a spatial map corresponding to different locations at the input. We can use this idea to efficiently train CONV-WTA autoencoders on datasets containing large images. Suppose we want to train an AlexNet [1] architecture in an unsupervised fashion on ImageNet, ILSVRC-2012

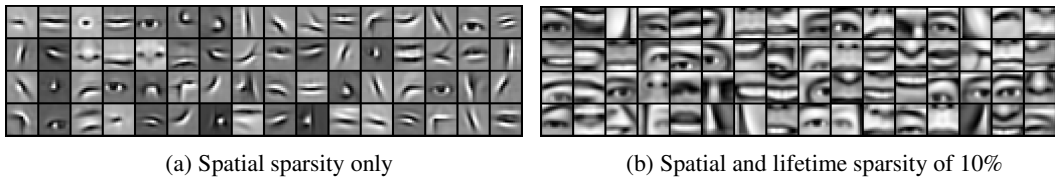

| (a) Spatial sparsity only | (b) Spatial and lifetime sparsity of 10% |

Figure 7: Learnt dictionary (deconvolution filters) of CONV-WTA autoencoder trained on the Toronto Face dataset (64conv7-64conv7-64conv7-64deconv15).

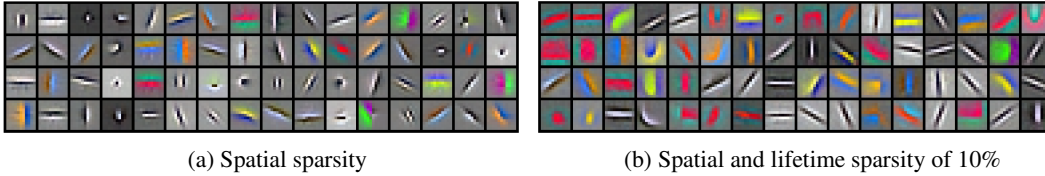

(a) Spatial sparsity                                    (b) Spatial and lifetime sparsity of 10%

Figure 8: Learnt dictionary (deconvolution filters) of CONV-WTA autoencoder trained on ImageNet $48 \times 48$ whitened patches. (64conv5-64conv5-64conv5-64deconv11).

(224x224). In order to learn the first layer $11 \times 11$ shift-invariant filters, we can extract medium-size image patches of size $48 \times 48$ and train a CONV-WTA autoencoder with 64 dictionary atoms of size 11 on these patches. This will result in 64 shift-invariant filters of size $11 \times 11$ that can efficiently capture the statistics of $48 \times 48$ patches. Once the filters are learnt, we can apply them in a convolutional fashion with the stride of 4 to the entire images and after max-pooling we will have a $64 \times 27 \times 27$ representation of the images. Now we can train another CONV-WTA autoencoder on top of these feature maps to capture the statistics of a larger receptive field at different location of the input image. This process could be repeated for multiple layers. Fig. 8 shows the dictionary learnt on the ImageNet using this approach. We can see that by imposing lifetime sparsity, we could learn very diverse filters such as corner, circular and blob detectors.

## 4 Experiments

In all the experiments of this section, we evaluate the quality of unsupervised features of WTA autoencoders by training a naive linear classifier (*i.e.*, SVM) on top them. We did not fine-tune the filters in any of the experiments. The implementation details of all the experiments are provided in Appendix A (*in the supplementary materials*). An IPython demo for reproducing important results of this paper is publicly available at `http://www.comm.utoronto.ca/~makhzani/`.

### 4.1 Winner-Take-All Autoencoders on MNIST

The MNIST dataset has 60K training points and 10K test points. Table 1 compares the performance of FC-WTA autoencoder and WTA-RBMs with other permutation-invariant architectures. Table 2a compares the performance of CONV-WTA autoencoder with other convolutional architectures. In these experiments, we have used all the available training labels ($N = 60000$ points) to train a linear SVM on top of the unsupervised features.

An advantage of unsupervised learning algorithms is the ability to use them in semi-supervised scenarios where labeled data is limited. Table 2b shows the semi-supervised performance of a CONV-WTA where we have assumed only $N$ labels are available. In this case, the unsupervised features are still trained on the whole dataset (60K points), but the SVM is trained only on the $N$ labeled points where $N$ varies from 300 to 60K. We compare this with the performance of a supervised deep convnet (CNN) [17] trained only on the $N$ labeled training points. We can see supervised deep learning techniques fail to learn good representations when labeled data is limited, whereas our WTA algorithm can extract useful features from the unlabeled data and achieve a better classification. We also compare our method with some of the best semi-supervised learning results recently obtained by

| | Error Rate |
|---|---|
| Shallow Denoising/Dropout Autoencoder (20% input and 50% hidden units dropout) | 1.60% |
| Stacked Denoising Autoencoder (3 layers) [9] | 1.28% |
| Deep Boltzmann Machines [13] | 0.95% |
| $k$-Sparse Autoencoder [14] | 1.35% |
| **Shallow FC-WTA Autoencoder, 2000 units, 5% sparsity** | 1.20% |
| **Stacked FC-WTA Autoencoder, 5% and 2% sparsity** | 1.11% |
| Restricted Boltzmann Machines | 1.60% |
| **Winner-Take-All Restricted Boltzmann Machines (30% sparsity)** | 1.38% |

Table 1: Classification performance of FC-WTA autoencoder features + SVM on MNIST.

| | Error |
|---|---|
| Deep Deconvolutional Network [7, 12] | 0.84% |
| Convolutional Deep Belief Network [5] | 0.82% |
| Scattering Convolution Network [15] | 0.43% |
| Convolutional Kernel Network [16] | 0.39% |
| **CONV-WTA Autoencoder, 16 maps** | 1.02% |
| **CONV-WTA Autoencoder, 128 maps** | 0.64% |
| **Stacked CONV-WTA, 128 & 2048 maps** | 0.48% |

| N | CNN [17] | CKN [16] | SC [15] | **CONV-WTA** |
|---|---|---|---|---|
| 300 | 7.18% | 4.15% | 4.70% | **3.47%** |
| 600 | 5.28% | - | - | **2.37%** |
| 1K | 3.21% | 2.05% | 2.30% | **1.92%** |
| 2K | 2.53% | 1.51% | **1.30%** | 1.45% |
| 5K | 1.52% | 1.21% | **1.03%** | 1.07% |
| 10K | **0.85%** | 0.88% | 0.88 % | 0.91% |
| 60K | 0.53% | **0.39%** | 0.43% | 0.48% |

(a) Unsupervised features + SVM trained on $N = 60000$ labels (no fine-tuning)

(b) Unsupervised features + SVM trained on few labels $N$. (semi-supervised)

Table 2: Classification performance of CONV-WTA autoencoder trained on MNIST.

convolutional kernel networks (CKN) [16] and convolutional scattering networks (SC) [15]. We see CONV-WTA outperforms both these methods when very few labels are available ($N < 1K$).

## 4.2 CONV-WTA Autoencoder on Street View House Numbers

The SVHN dataset has about 600K training points and 26K test points. Table 3 reports the classification results of CONV-WTA autoencoder on this dataset. We first trained a shallow and stacked CONV-WTA on all 600K training cases to learn the unsupervised features, and then performed two sets of experiments. In the first experiment, we used all the N=600K available labels to train an SVM on top of the CONV-WTA features, and compared the result with convolutional $k$-means [11]. We see that the stacked CONV-WTA achieves a dramatic improvement over the shallow CONV-WTA as well as $k$-means. In the second experiment, we trained an SVM by using only $N = 1000$ labeled data points and compared the result with deep variational autoencoders [18] trained in a same semi-supervised fashion. Fig. 9 shows the learnt dictionary of CONV-WTA on this dataset.

| | Accuracy |
|---|---|
| Convolutional Triangle $k$-means [11] | 90.6% |
| **CONV-WTA Autoencoder, 256 maps (N=600K)** | 88.5% |
| **Stacked CONV-WTA Autoencoder, 256 and 1024 maps (N=600K)** | 93.1% |
| Deep Variational Autoencoders (non-convolutional) [18] (N=1000) | 63.9% |
| **Stacked CONV-WTA Autoencoder, 256 and 1024 maps (N=1000)** | 76.2% |
| *Supervised* Maxout Network [19] (N=600K) | 97.5% |

Table 3: CONV-WTA unsupervised features + SVM trained on $N$ labeled points of SVHN dataset.

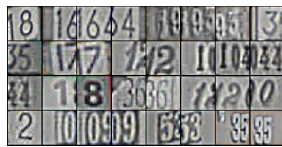

(a) Contrast Normalized SVHN

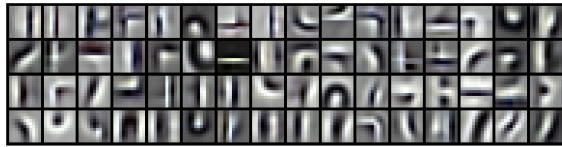

(b) Learnt Dictionary (64conv5-64conv5-64conv5-64deconv11)

Figure 9: CONV-WTA autoencoder trained on the Street View House Numbers (SVHN) dataset.

## 4.3 CONV-WTA Autoencoder on CIFAR-10

Fig. 10a reports the classification results of CONV-WTA on CIFAR-10. We see when a small number of feature maps ($< 256$) are used, considerable improvements over $k$-means can be achieved. This is because our method can learn a shift-invariant dictionary as opposed to the redundant dictionaries learnt by patch-based methods such as $k$-means. In the largest deep network that we trained, we used 256, 1024, 4096 maps and achieved the classification rate of 80.1% without using fine-tuning, model averaging or data augmentation. Fig. 10b shows the learnt dictionary on the CIFAR-10 dataset. We can see that the network has learnt diverse shift-invariant filters such as point/corner detectors as opposed to Fig. 2b that shows the position-specific filters of patch-based methods.

| | Accuracy |
|---|---|
| Shallow Convolutional Triangle *k*-means (64 maps) [3] | 62.3% |
| **Shallow CONV-WTA Autoencoder (64 maps)** | 68.9% |
| Shallow Convolutional Triangle *k*-means (256 maps) [3] | 70.2% |
| **Shallow CONV-WTA Autoencoder (256 maps)** | 72.3% |
| Shallow Convolutional Triangle *k*-means (4000 maps) [3] | 79.6% |
| Deep Triangle *k*-means (1600, 3200, 3200 maps) [20] | 82.0% |
| Convolutional Deep Belief Net (2 layers) [6] | 78.9% |
| Exemplar CNN (300x Data Augmentation) [21] | 82.0% |
| NOMP (3200,6400,6400 maps + Averaging 7 Models) [22] | 82.9% |
| **Stacked CONV-WTA (256, 1024 maps)** | 77.9% |
| **Stacked CONV-WTA (256, 1024, 4096 maps)** | 80.1% |
| *Supervised* Maxout Network [19] | 88.3% |

(a) Unsupervised features + SVM (without fine-tuning)

(b) Learnt dictionary (deconv-filters)
64conv5-64conv5-64conv5-64deconv7

Figure 10: CONV-WTA autoencoder trained on the CIFAR-10 dataset.

## 5 Discussion

**Relationship of FC-WTA to *k*-sparse autoencoders.** *k*-sparse autoencoders impose sparsity across different channels (population sparsity), whereas FC-WTA autoencoder imposes sparsity across training examples (lifetime sparsity). When aiming for low sparsity levels, *k*-sparse autoencoders use a scheduling technique to avoid the dead dictionary atom problem. WTA autoencoders, however, do not have this problem since all the hidden units get updated upon visiting every mini-batch no matter how aggressive the sparsity rate is (no scheduling required). As a result, we can train larger networks and achieve better classification rates.

**Relationship of CONV-WTA to deconvolutional networks and convolutional PSD.** Deconvolutional networks [7, 12] are top down models with no direct link from the image to the feature maps. The inference of the sparse maps requires solving the iterative ISTA algorithm, which is costly. Convolutional PSD [4] addresses this problem by training a parameterized encoder separately to explicitly predict the sparse codes using a soft thresholding operator. Deconvolutional networks and convolutional PSD can be viewed as the generative decoder and encoder paths of a convolutional autoencoder. Our contribution is to propose a specific winner-take-all approach for training a convolutional autoencoder, in which both paths are trained jointly using direct backpropagation yielding an algorithm that is much faster, easier to implement and can train much larger networks.

**Relationship to maxout networks.** Maxout networks [19] take the max across different channels, whereas our method takes the max across space and mini-batch dimensions. Also the winner-take-all feature maps retain the location information of the "winners" within each feature map and different locations have different connectivity on the subsequent layers, whereas the maxout activity is passed to the next layer using weights that are the same regardless of which unit gave the maximum.

## 6 Conclusion

We proposed the winner-take-all spatial and lifetime sparsity methods to train autoencoders that learn to do fully-connected and convolutional sparse coding. We observed that CONV-WTA autoencoders learn shift-invariant and diverse dictionary atoms as opposed to position-specific Gabor-like atoms that are typically learnt by conventional sparse coding methods. Unlike related approaches, such as deconvolutional networks and convolutional PSD, our method jointly trains the encoder and decoder paths by direct back-propagation, and does not require an iterative EM-like optimization technique during training. We described how our method can be scaled to large datasets such as ImageNet and showed the necessity of the deep architecture to achieve better results. We performed experiments on the MNIST, SVHN and CIFAR-10 datasets and showed that the classification rates of winner-take-all autoencoders are competitive with the state-of-the-art.

**Acknowledgments**

We would like to thank Ruslan Salakhutdinov and Andrew Delong for the valuable comments. We also acknowledge the support of NVIDIA with the donation of the GPUs used for this research.

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
