[Supplementary Material · appendix.pdf]

# Appendix A   Implementation Details

In this section, we describe the network architectures and hyper-parameters that were used in the experiments. While most of the conventional sparse coding algorithms require complex matrix operations such as matrix inversion or SVD decomposition, WTA autoencoders only require the *sort* operation in addition to matrix multiplication and convolution which are all efficiently implemented in most GPU libraries. We used Alex Krizhevsky's *cuda-convnet* convolution kernels [1] for this work.

## A.1   Deconvolution Kernels

At the decoder of a convolutional autoencoder, deconvolutional layers are used. The deconvolution operation is exactly the reverse of convolution (*i.e.*, its forward pass is the backward pass of convolution). For example, whereas a strided convolution decreases the feature map size, a strided deconvolution increases the map size. We implemented the deconvolution kernels by minor modifications of current available GPU kernels for the convolution operation.

## A.2   Effect of Tied Weights

We found that tying the encoder and decoder of FC-WTA autoencoders helps the generalization performance of them. But tying the convolution and deconvolution weights of WTA-CONV autoencoders hurts the generalization performance (data not shown). We think this is because the CONV-WTA autoencoder is already very regularized by the aggressive sparsity constraints and tying the weights results in too much regularization.

## A.3   WTA-CONV Autoencoder on MNIST

On the MNIST dataset, we trained two networks:

**Shallow CONV-WTA Autoencoder (128 maps).** In the shallow architecture, we used 128 filters with a $7 \times 7$ receptive field applied at strides of 1 pixel. After training, we used max-pooling over $5 \times 5$ regions at strides of 3 pixels to obtain the final $128 \times 10 \times 10$ representation. SVM was then applied to this representation for classification.

**Stacked CONV-WTA Autoencoder (128, 2048 maps).** In the deep architecture, we trained another 2048 feature maps on top of the pooled feature maps of the first network, with a filter width of 3 applied at strides of 1 pixel. After training, we used max pooling over $3 \times 3$ regions at strides of 2 pixels to obtain the final $2048 \times 5 \times 5$ representation. SVM was then applied to this representation for classification.

**Semi-Supervised CONV-WTA Autoencoder.** In the semi-supervised setup, the amount of labeled data was varied from $N = 300$ to $N = 60000$. We ensured the dataset is balanced and each class has the same number of labeled points in all the experiments. We used the stacked CONV-WTA autoencoder (128, 2048 maps) trained in the previous part, and trained an SVM on top of the unsupervised features using only $N$ labeled data.

## A.4   WTA-CONV Autoencoder on SVHN

The Street View House Numbers (SVHN) dataset consists of about 600,000 images (both the difficult and the simple sets) and 26,000 test images. We first apply global contrast normalization to the images and then used local contrast normalization using a Gaussian kernel to preprocess each channel of the images. This is the same preprocessing that is used in [19]. The contrast normalized SVHN images are shown in Fig. 9b. We trained two networks on this dataset.

**CONV-WTA Autoencoder (256 maps).** The architecture used for this network is 256conv3-256conv3-256conv3-256deconv7. After training, we used max-pooling on the last 256 feature maps of the encoder, over $6 \times 6$ regions at strides of 4 pixels to obtain the final $256 \times 8 \times 8$ representation. SVM was then applied to this representation for classification. We observed that having a stack of conv3 layers instead of a 256conv7 encoder, significantly improved the classification rate.

**Stacked CONV-WTA Autoencoder (256, 1024 maps).** In the stacked architecture, we trained another 1024 feature maps on top of the pooled feature maps of the first network, with a filter width of 3 applied at strides of 1 pixel. After training, we used max pooling over $3 \times 3$ regions at strides of 2 pixels to obtain the final $1024 \times 4 \times 4$ representation. SVM was then applied to this representation for classification.

**Semi-Supervised CONV-WTA Autoencoder.** In the semi-supervised setup, we assumed only N=1000 labeled data is available. We used the stacked CONV-WTA autoencoder (256, 1024 maps) trained in the previous part, and trained an SVM on top of the unsupervised features using only $N = 1000$ labeled data.

## A.5   WTA-CONV Autoencoder on CIFAR-10

On the CIFAR-10 dataset, we used global contrast normalization followed by ZCA whitening with the regularization bias of 0.1 to preprocess the dataset. This is the same preprocessing that is used in [3]. We trained three networks on CIFAR-10.

**CONV-WTA Autoencoder (256 maps).** The architecture used for this network is 256conv3-256conv3-256conv3-256deconv7. After training, we used max-pooling on the last 256 feature maps of the encoder, over $6 \times 6$ regions at strides of 4 pixels to obtain the final $256 \times 8 \times 8$ representation. SVM was then applied to this representation for classification.

**Stacked CONV-WTA Autoencoder (256, 1024 maps).** For this network, we trained another 1024 feature maps on top of the pooled feature maps of the first network, with a filter width of 3 applied at strides of 1 pixel. After training, we used max pooling over $3 \times 3$ regions at strides of 2 pixels to obtain the final $1024 \times 4 \times 4$ representation. SVM was then applied to this representation for classification.

**Stacked CONV-WTA Autoencoder (256, 1024, 4096 maps).** For this model, we first trained a CONV-WTA network with the architecture of 256conv3-256conv3-256conv3-256deconv7. After training, we used max pooling on the last 256 feature maps of the encoder, over $3 \times 3$ regions at strides of 2 pixels to obtain a $256 \times 16 \times 16$ representation. We then trained another 1024 feature maps with filter width of 3 and stride of 1 on top of the pooled feature maps of the first layer. We then obtained the second layer representation by max pooling the 1024 feature maps with a pooling stride of 2 and width of 3 to obtain a $1024 \times 8 \times 8$ representation. We then trained another 4096 feature maps with filter width of 3 and the stride of 1 on top of the pooled feature maps of the second layer. Then we used max-pooling on the 4096 feature maps with a pooling width of 3 applied at strides of 2 pixels to obtain the final $4096 \times 4 \times 4$ representation. An SVM was trained on top of the final representation for classification.