[Reviews · NeurIPS 2015]

Submitted by Assigned_Reviewer_1

The paper proposes a novel way to train a sparse autoencoder where the hidden unit sparsity is governed by a winner-take-all kind of selection scheme. This is a convincing way to achieve a sparse autoencoder, while the paper could have included some more details about their training strategy and the complexity of the algorithm.

I think the main claim that the winner-take-all encoding can give the desired sparsity in a convenient way, is convincing. And, the paper indeed provides a thorough investigation about the possible options especially in the sense of the use in a CNN setting and in the unsupervised feature learning context.

What I think the paper is missing are two parts: the effect of fine-tuning and the complexity of the winner-take-all encoding. First, the paper discusses only the greedy layer-wise unsupervised learning option in order to construct a stacked WTA autoencoders. I wonder why the authors didn't go ahead and try to fine-tune their stacked autoencoders to see if the network performance is still good enough when there are labeled data samples available. It might be also interesting information for the readers if they have included some results from training a multi-layer network directly coupled with or without dropout technique.

Another thing is that there's a lack of discussion about the complexity of finding those k winners. I presume that this complexity must not be very high, because their mini-batch-based processing, but usually this can involve frequent sorting operations during the feedforward. It could be nicer to include those analysis.
Summary: The paper proposes a novel way to train a sparse autoencoder where the hidden unit sparsity is governed by a winner-take-all kind of selection scheme. This is a convincing way to achieve a sparse autoencoder, while the paper could have included some more details about their training strategy and the complexity of the algorithm.

Submitted by Assigned_Reviewer_2

The paper describes a new scheme, based on a winner take all idea for learning auto-encoders. Extensive comparisons with alternative methods are provided.

Pro: new learning scheme that can be used for both deterministic and stochastic stacked auto-encoders. The idea is competitive with the existing literature.

Cons: one more method for learning with auto-encoders.
Summary: aa

Submitted by Assigned_Reviewer_3

The authors present a fully connected auto-encoder with a new sparsity constraint called the lifetime sparsity. For each hidden unit across the mini-batch, they rank the activation values, keeping only the top-k% for reconstruction. The approach is appealing because they don't need to find a hard threshold and it makes sure every hidden unit/filter is updated (no dead filters because their activation was below the threshold).

Their encoder is a deep stack of ReLu and the decoder is shallow and linear (note that usually non-symmetric auto-encoders lead to worse results). They also show how to apply to RBM. The effect of sparsity is very effective and noticeable on the images depicting the filters.

They extend this auto-encoder in a convolutional/deconvolutional framework, making it possible to train on larger images than MNIST or TFD. They add a spatial sparsity, keeping the top activation per feature map for the reconstruction and combine it with the lifetime sparsity presented before.

Having worked on auto-encoders in the past, I really like the paper and I think this will be very interesting to many people in this field (researchers in auto-encoder representation learning). I have a few questions/requests:

* can you present the reconstruction of a full image on image net using your architecture? i.e. presenting the original image on one side and the reconstructed version on the side. * you only evaluate with an SVM on top of the representation. Does the pre-initialization improves the performance of the same architecture but randomly initialized if you fine-tune the model? can you provide results in that scenario?

minor comments L52 "no hyper-parameter to be tuned" what about the initialization technique you choose? the learning rate? the number of hidden units? the activation function? I would consider those as hyper-parameters. L230 "it requires all the dictionary atoms to contribute in the reconstruction" if I look at Fig 5f, there are a few feature maps that are totally black (meaning no activation at all, or very close to 0). Will the corresponding atoms contribute to the reconstruction? same idea at L236, "every filter will get updated", maybe precise that the filter needs to be activated. Maybe you can measure during training which proportion of the filters are not activated. L295 can you point to the url if the paper gets accepted? thanks. Table 2b, CONV-WTA = Stacked CONV-WTA 128 & 2048 maps? precise in the caption. Fig 4a, filters look very weird, can you explain?

precise how ranking at each iteration affect the training time (should be very negligible but useful to state that it does compare to normal auto encoder training)
Summary: Having worked on auto-encoders in the past, I really enjoy reading this paper and I think this will be very interesting to many people in this field (researchers in auto-encoder representation learning).

Submitted by Assigned_Reviewer_4

The paper presents a strategy for learning sparse representation in an unsupervised way. While a number of (often computer vision) tasks are tackled using huge labeled datasets, being able to learn relevant representations in an unsupervised way is a key issue for a number of settings.

The proposed approach exploits on a mechanism close to the one of k-sparse autoencoders proposed by Makkhzani et al [14]. The authors extend the idea from [14] to build winner-take-all encoders (and RBMs), that enforce both spatial and lifetime regularization by keeping only a percentage (the biggest) of activations. The lifetime sparsity allows overcoming problems that could arise with k-sparse autoencoders. The authors next propose to embed their modeling framework in convolutional neural nets to deal with larger images than e.g. those of mnist.

As demonstrated by the experimental results (both the illustration of learnt filters and classification experiments with svm exploiting the learnt representations) the obtained representations seem to be much relevant and perform reasonably well with respect to fully supervised methods, especially with few labeled training data.

Overall the paper is well written and understandable. The ideas are not revolutionary but these are sounded and seem to provide good experimental results on three standard benchmark datasets.

few comments: - The beginning of part 3 is too long in my opinion to deliver the simple message that convolution is good for images.

- i don't understand what you mean page 3 in the sentence : "Our work i similar to deconvolutional... for directly learning a sparse autoencoder". could you reformulate this ? - section 3.1 : Why do you use such a hard sparsification (keeping only one activity) ? did you try using more activities (a percentage) ?

- how do you build the images for deconvolutional filters in fig 5 b? - end of section 3.4: it looks like you could learn a convolutional WTA model directly without going through the step of learning a Conv-WTA on 48x48 patches ? What is the advantage of this strategy ? - How do the methods in table 2 compare wrt. the complexity (number of parameters) ?
Summary: The paper presents a strategy for learning sparse representation in an unsupervised way. The proposed approach exploits on a mechanism for enforcing sparse representation in convolutional networks.

Overall the paper is well written and understandable. The ideas are not revolutionary but these are sounded and seem to provide good experimental results on three standard benchmark datasets.

Author Feedback
Author rebuttal: We thank the reviewers for their feedback.

1. R1 and R2 ask about the effect of fine-tuning and a comparison of pre-trained vs randomly initialized networks. When limited labels are available (see Table 2b), our CONV-WTA, even without fine-tuning, outperforms a randomly initialized supervised network (CNN). Fine-tuning in this case only slightly improved results. When a larger number of labels are available, fine-tuning improved our results, but only marginally, compared to a randomly initialized supervised network with dropout. However, we observed that a pre-trained network was able to match the state-of-the-art much faster than a supervised dropout network trained from scratch. In short, we believe the main advantage of our method is the ability to use it in semi-supervised scenarios where labeled data is limited. We will include a full discussion of the fine-tuning experiments in the final manuscript.

2. R1, R2 and R3 ask about the complexity of winner-take-all encoding, the details of training strategy and the code url. We have released the code as well as an ipython demo replicating the important results in the paper. We will include the url in the final paper. The training details can be found both in the submitted appendix and the released code. The complexity of sorting is NBlogB, where B is the mini-batch size (100) and N is the number of hidden units. This is negligible compared to the matrix multiplication complexity. Comprehensive profiling of our code indicates that most time is spent in the multiplication/convolution kernels; the mini-batch sorting operation only marginally slows down the network.

3. R2: minor comments:
- We will include the reconstructed image in the final paper. The reconstruction error is visually negligible.
- Fig 4a is discussed in L149.
- Table 2: CONV-WTA = Stacked CONV-WTA

4. R3: "Why do you use such a hard sparsification"? In order to address this question, we did an experiment where the top k activities are kept, not just the top activity. Higher k gives the model more capacity and reduces the reconstruction error, but it also increases the classification error by preventing the model from learning sharp filters. We will include these results in the final paper.

5. R3: "How do the methods in Table 2 compare wrt complexity?" Deconvolutional networks use expensive EM-like ISTA operations and DBNs use contrastive divergence. Our method, however, uses direct backpropagation which is faster and enables us to train larger networks. Also, CONV-WTA does not require more capacity. For example, a deconvnet [12] with 16 maps achieves 1.38% on MNIST while our method with 16 maps and exact same arch. of deconvnet at the decoder, achieves 1.02% error (see Table 2a).

6. R3: minor comments:
- The deconv operation (explained in Appendix), maps each input hidden unit to an output 11x11 image patch, which can be visualized.
- We trained the CONV-WTA on 48x48 patches instead of the whole image, only to reduce the amount of GPU memory needed and speed up learning.

7. R5: "cons: one more method for learning with auto-encoders." Yes, ours is another method for learning autoencoders, but ours works significantly better than the state of the art, including variational, denoising and dropout autoencoders, and deconv. networks.

8. R6: "the novelty of the method is limited as the sparsity, has been used in numerous papers in similar settings". It is true that sparsity is a widely studied topic, but our paper significantly advances the field by introducing a quite different sparsity constraint that performs better than all state-of-the-art sparsity methods. Further, it trains very fast (~10 epochs) and is much simpler to implement. We cut the state-of-the-art error for sparse unsupervised MNIST nearly in half (0.48% error versus 0.84% for deconvnets, 0.82% conv. RBM), and we perform better than several complicated models on SVHN and CIFAR-10. Previously, learning sparse unsupervised convolutional features has mostly been done via EM-like algorithms (conv. PSD, deconvnets) and sometimes via contrastive divergence (conv. RBM). The novelty of our work is proposing a direct backpropagation algorithm for sparse representation learning which proves to be more effective and yet much faster.

9. R6 raises concerns about the motivation of lifetime sparsity. We introduced lifetime sparsity to help learn rare but useful features, by preventing filters from dying or from being used for every image; both situations produce uninformative filters. Imposing fixed lifetime sparsity helps find the right balance. We have shown the effect of our WTA life-time sparsity by depicting the filters (in Figs. 1,6,7,8) and showing classification results. Lines 81 to 87 specifically discuss the effect of lifetime sparsity in fully-connected architecture and Section 3.2 discusses its effect in convolutional settings. We will modify the introduction to better explain the motivation.